# Patient priorities in relation to surgery for gastric cancer: qualitative interviews with gastric cancer surgery patients to inform the development of a core outcome set

Bilal Alkhaffaf [iD] ,[1,2] Jane M Blazeby,[3] Iain A Bruce,[4,5] Rebecca L Morris [iD] [6]

For numbered affiliations see end of article.

**Correspondence to**
Mr Bilal Alkhaffaf;
bilal.alkhaffaf@srft.nhs.uk

## ABSTRACT

**Objective** The reporting of outcomes in surgical trials for gastric cancer is inconsistent. The GASTROS study (**GA**stric Cancer **S**urgery **TR**ials **R**eported **O**utcome **S**tandardisation) aims to address this by developing a core outcome set (COS) for use in all future trials within this field. A COS should reflect the views of all stakeholders, including patients. We undertook a series of interviews to identify outcomes important to patients which would be considered for inclusion in a COS.

**Setting** All interviews took place within the UK. Interviews were carried out face-to-face at hospitals and cancer support centres or via the telephone.

**Participants** Twenty participants at varying stages of recovery following surgery for gastric cancer with curative intent.

**Design** Qualitative design using semistructured interviews, supported by an interview guide which was iteratively modified; thematic analysis was used to explore patient priorities.

**Results** Six themes enveloping 38 outcomes were identified; surviving and controlling cancer, technical aspects of surgery, adverse events from surgery, recovering from surgery, long-term problems following surgery and long-term life impact of surgery. The 'most important' patient priority was to be 'cured of cancer'.

**Conclusion** Surgical trials for gastric cancer should consider broader priorities of patients when choosing which outcomes to report. This study highlighted the importance of longer-term outcomes such as cancer survival. Outcomes identified in this study will be used to inform an international Delphi survey to develop a COS in this field.

## INTRODUCTION
### Background

Gastric cancer is a leading cause of cancer-related death worldwide.[1 2] While surgery remains the only treatment which can offer a potential cure from this disease, it is associated with significant rates of morbidity in both the short and the long term.[3 4] Ideally, the optimal surgical approach would minimise

### Strengths and limitations of this study

► This is the first in-depth qualitative study to examine the priorities and experiences of patients following potentially curative surgery for gastric cancer, and the first to establish which outcomes are important to patients.

► The study forms part of a larger project (The GASTROS Study—www.gastrosstudy.org) to develop a 'core outcome set' for use in surgical trials for gastric cancer and was reviewed and funded by the National Institute of Health Research (UK). The study is based on a reproducible and transparent methodology which has been subjected to critical appraisal during a peer-review process.

► The term 'outcome' was described to participants in a manner relatable to them, such that they understood it and were able to identify which outcomes were most important.

► The patient population was limited to UK-based English speakers. The views of international patients may vary due to differences in culture and clinical practice.

the risk of short-term and long-term complications without jeopardising the oncological resection.

Identifying the optimal surgical approach for gastric cancer should be based on comparing and combining robust clinical evidence from well-designed randomised controlled trials. One of the present challenges to achieving this is the inconsistency in the reporting of outcomes in this field.[5] This limits evidence synthesis and contributes to 'research waste'. The GASTROS study (GASTROS—**GA**stric Cancer **S**urgery **TR**ials **R**eported **O**utcome **S**tandardisation—www.gastrosstudy.org)[6] aims to address this issue by developing a 'core outcome set' (COS)—a *minimum* group of standardised

and well-defined outcomes, measured by all future gastric cancer surgery trials.[7]

A guiding principle in the development of COS is that outcomes reflect the views and priorities of key stakeholders, including patients, to maximise the relevance and impact of future research. Previous studies have demonstrated variations in the views and priorities of clinicians and patients,[8–10] which can result in trials reporting outcomes which bear little relevance to patients. A systematic review of outcome reporting in surgical trials for gastric cancer has demonstrated that outcomes which may be important to patients with gastric cancer, such as 'quality of life' after surgery, are poorly represented within this field.[5] It is therefore important to understand which outcomes are important for patients undergoing gastric cancer surgery.

## Objective

This research forms part of the GASTROS study, for which the protocol has been previously described.[6] The first stage in the study involves identifying a 'long-list' of potentially important outcomes which will be prioritised in stage two by participants undertaking a Delphi survey. It is not known to what degree outcomes reported in previously published trials represent the priorities of patients undergoing gastric cancer surgery, and as such, solely relying on these as a source to populate the 'long-list' may overlook potentially important outcomes. By exploring the experiences, perceptions and priorities of patients who have undergone surgery for gastric cancer, this study aimed to identify outcomes which may not have been previously reported in the literature.

## METHODS
### Study design

The role of qualitative research methods in the development of COS has been previously explored[11] and has been advocated by groups such as the Core Outcome Measures in Effectiveness Trials (COMET) initiative[7] as one of several approaches to ensure that outcome lists being considered for prioritisation are exhaustive. This qualitative study used a semistructured interview approach to achieve the primary objective of identifying outcomes of importance to patients. A series of open questions were used to facilitate a patient-led discussion, guided by additional prompts from a pre-prepared interview schedule (table 1) to ensure key areas were covered.

Additional focused questioning around the use of outcomes in research was also included. In the context of clinical research, terms such as 'outcomes' may not be well understood by patients[11] and so a mixture of open and closed questioning was important. Participant interviews were undertaken in series of three, following which transcript analysis (see below) was undertaken and the interview schedule was modified iteratively. This ensured that areas raised by earlier participants, but not included

in the original schedule, were covered in subsequent discussions.

### Sampling

The eligibility criteria for this study are summarised in table 2. A purposive sampling strategy was adopted across the following characteristics:

▶ Age (above and below 70 years).
▶ Gender (men and women).
▶ Time since surgery (less than a year, 1–3 years and more than 3 years).

Interviews were undertaken until 'data-saturation' was achieved. Data saturation was determined when there was no new data emerging that had interpretive value. (table 2)

Participants were recruited from across the UK from three sources:

1. A regional specialist gastric cancer centre: patients were approached in the outpatient clinic by their direct care team.
2. Patient organisations: patient groups were asked to contact their membership through email and social media.
3. Snowball sampling; patients who had been recruited or contacted to participate were asked to identify other patients who would be interested in the study.

### Data collection

Interviews were undertaken between February and May 2017 and were conducted by BA, a consultant surgeon and researcher with approximately 10 years' experience of managing and communicating with patients with gastric cancer. Participants were invited to choose between a University Teaching Hospital, two purpose-built patient cancer centres and their home for the location of the meeting. Participants were also offered the opportunity to have their interviews over the telephone. Participants were offered travel expenses to minimise any financial burden on taking part in the study. In addition to the purposive sampling strategy, the following demographic data were collected:

▶ Gender (male/female).
▶ Social circumstances (eg, lives alone/with partner/ lives with dependents).
▶ Age.
▶ Time since surgery (in months).
▶ The type of gastrectomy (total or partial gastrectomy).
▶ The approach to their surgery (open or laparoscopic).
▶ Whether they had undergone additional treatment (eg, chemotherapy).
▶ Whether they had suffered a postoperative complication.
▶ Ethnicity.
▶ Previous trial enrolment.
▶ Participant's postcode (to identify location and social deprivation score).

Full written consent was taken immediately prior to the interview and the participants were reminded that they

| Table 1 | Interview schedule |
|---|---|
| 1. | I understand you have (had) gastric cancer. Can you tell me about that? |
| 2. | *Could you tell me about how you first found out you had gastric cancer?*<br>Prompts:<br>▸ *What questions did you most want to ask, when you were told that you had gastric cancer?* |
| 3 | Were there were any areas you wanted more information about but were unable to find?<br>Prompts:<br>▸ *Were you given any leaflets at the time of diagnosis? Did you find these useful?* |
| 4 | What treatment was offered and how you decide about undergoing treatment.<br>Prompts:<br>▸ *What information did you want about the treatment you would be receiving?*<br>▸ *What factors did you consider when deciding on the treatment?'* |
| 5 | What effects did the treatment have on you after surgery?<br>Prompts:<br>▸ *Did the treatment affect your physical or mental well-being?*<br>▸ *Did the treatment have an effect on relationships with those around you?*<br>▸ *Did you have to make any changes to your behaviour as a result of treatment?* |
| 6 | What long-term effects did the treatment have on you?<br>Prompts:<br>▸ *Did the treatment affect your physical or mental well-being?*<br>▸ *Did the treatment have an effect on relationships with those around you?*<br>▸ *Did you have to make any changes to your behaviour as a result of treatment?* |
| 7 | What was the worst side effect of treatment? |
| 8 | What are your concerns for the future, especially those relating to their diagnosis/history of gastric cancer? |
| 9 | Did the explanation of what you should expect from surgery match your real experience? |
| 10 | In the context of research studies, can you explain what an outcome is in your own words?<br>The interviewer will then provide a definition of the term 'outcome' is in the context of clinical research. |
| 11 | What, in your opinion, is the most important outcome to measure in gastric cancer surgery trials? |
| 12 | Are there any other outcomes which may be important to measure? |
| 13 | Has your perspective on what is important changed over time? |
| 14 | Is there anything else that you feel is important to talk about that we have not discussed? |

can stop at any point or withdraw from the study without needing to give a reason.

### Data analysis

All interviews were digitally recorded and professionally transcribed (intelligent verbatim transcription). A thematic analysis was used to identify emerging themes and was guided by a general inductive approach.[12 13] This was used to create the framework applied to subsequent interviews. Themes were developed using a three-step approach of open coding, axial coding and selective coding[14] of the transcripts. Given the objective of this study was to identify themes and outcomes not previously

| Table 2 | Eligibility criteria for study participants | |
|---|---|---|
| | **Potential participants approached** | **Potential participants not approached** |
| Participant | ▸ Male and female aged 18 years and older.<br>▸ Individuals able to participate in an interview in the English language. | ▸ Patients unable to give informed consent<br>▸ Patients too unwell to comfortably participate in an interview lasting approximately 30–60 min. |
| Pathology | ▸ Adenocarcinoma and squamous cell carcinoma of the stomach (which makes up 95% of all stomach tumours). | ▸ Gastrointestinal stromal tumours<br>▸ Neuroendocrine tumours<br>▸ Lymphoma<br>▸ Benign disease |
| Intervention | ▸ Total and partial gastrectomy<br>▸ Open and laparoscopic approaches | ▸ Surgery with palliative intent<br>▸ Endoscopic therapies such as EMR (endoscopic mucosal resection) and ESD (endoscopic submucosal dissection) |

reported in trials, it was important not to base data analysis and outcome identification on a framework built on previously published literature. BA and RLM (a researcher with significant experience in qualitative research methods) independently analysed the first two transcripts and through discussion identified themes and adjustments to the interview schedule. There were no disagreements about coding, but had there been, these would have been discussed with the study management team. The final themes were agreed by all authors through discussion. Data analysis was supported using NVivo V.11 (http://www.qsrinternational.com/products_nvivo.aspx, QSR International, Burlington, Massachusetts, USA).

### Portfolio adoption

The study was adopted by the National Institute for Health Research (NIHR) Clinical Research Network Portfolio (ID 33312).

### Reporting

This paper uses the SRQR checklist to structure the report of the study findings.[15]

### Patient and public involvement

A Study Advisory Group (SAG) forms part of the management structure of the wider GASTROS study,[6] of which this qualitative study forms part of the first stage. The SAG is made up of key stakeholder representatives including patients, oncology nurses and surgeons. The group provides advice on the methodology of the study, general delivery of the study against its stated objectives and ensures that the viewpoints of all stakeholder groups are considered. The results of this study were presented to a SAG meeting; the ensuing discussion influenced the design of the next stage of the study in preparation for an an international Delphi survey.

Best practice guidelines for patient and public engagement were followed as set out by INVOLVE (part of and funded by the UK's National Institute for Health Research).[16]

## RESULTS
### Overview

In total, 20 patients were interviewed. Table 3 summarises demographic data and treatment-related characteristics of participants. Interviews lasted a median of 50.5 min (29–75 min). No patients withdrew from the study. Data saturation was deemed to have been reached by 20 interviews; one new outcome was identified in interview number 18 (related to sexual activity); however, no further outcomes were identified from the following two transcripts.

### Outcome themes

Six broad themes enveloping 38 outcomes were identified:
1. Surviving and controlling cancer.
2. Technical aspects of surgery.
3. Adverse events from surgery.
4. Recovering from surgery.
5. Long-term problems following surgery.
6. Long-term life impact of surgery.

Online supplementary appendix 1 provides a breakdown of how all themes were developed from outcomes identified during the interviews. Themes were well represented in each interview; each theme was discussed by at least 18 of 20 participants. Online supplementary appendix 2 demonstrates the outcomes identified during the interviews and how often they were referenced.

### Theme 1: surviving and controlling cancer

For most, details of their initial consultation were sketchy; participants often described being given lots of information about their diagnosis, much of which was not absorbed. However, patients clearly remember their reaction to being told their cancer diagnosis; for most, the response was the same:

> When you hear the word, cancer, you think that's it. I'm going to die. (Participant 6)

There was a range of personal experience with cancer within our patient group. Some had direct family members who had undergone chemotherapy and had an intimate knowledge of its effects.

> It was my worst nightmare come true because I lost my dad to cancer and I always had it in the back of my mind, well if one person in the family could get cancer from somewhere then we could as well. (Participant 19)

Some patients had a vague knowledge of friends or work colleagues who had undergone treatments for cancer and others had no prior experience of cancer at all. Despite these differences, the initial responses to their diagnosis were similar.

All participants in our study had undergone radical surgery with curative intent. At the time of interview, no participants had confirmed evidence of disease recurrence, although one was being investigated for potential recurrence. Once the discussion with their surgeon moved away from the diagnosis and onto potentially curative treatments, participants often focused their questions on 'survival':

> I wanted to know what the chances were of me having this removed and not, well, basically not dying from it. (Participant 4)

Despite radical surgery (and perioperative chemotherapy in half of our participants), for many of the participants the fear of recurrence remained a permanent anxiety. Many participants seemed to understand that due to the aggressive nature of gastric cancer, recurrence is a possibility for many:

> …you're always worried that it's going to come back… (Participant 7)

**Table 3** Patient characteristics and demographic data

| Patient | Sex | Age | Ethnicity* | Social deprivation quintile† | Home circumstances | Months since surgery | Type of surgery | Approach to surgery | Postoperative complications | Perioperative treatment |
|---|---|---|---|---|---|---|---|---|---|---|
| 1 | F | 74 | A | 3rd | Lives alone | 15 | Partial gastrectomy | Laparoscopic | Yes | Nil |
| 2 | M | 59 | B | 3rd | Lives alone | 27 | Total gastrectomy | Open surgery | Yes | Chemotherapy |
| 3 | M | 71 | A | 1st | Lives alone | 16 | Partial gastrectomy | Open surgery | Yes | Nil |
| 4 | M | 43 | A | 2nd | Lives with parents | 15 | Total gastrectomy | Open surgery | Yes | Chemotherapy |
| 5 | M | 80 | A | 3rd | Lives alone | 23 | Partial gastrectomy | Laparoscopic | Yes | Nil |
| 6 | F | 52 | A | 2nd | Lives with children | 32 | Total gastrectomy | Open surgery | No | Chemotherapy |
| 7 | M | 79 | A | 1st | Lives with spouse | 58 | Total gastrectomy | Laparoscopic | Yes | Chemotherapy |
| 8 | F | 63 | A | 1st | Lives alone | 5 | Total gastrectomy | Open surgery | No | Nil |
| 9 | M | 61 | A | 3rd | Lives with spouse | 170 | Total gastrectomy | Open surgery | No | Nil |
| 10 | M | 61 | C | 1st | Lives alone | 79 | Total gastrectomy | Open surgery | No | Chemotherapy |
| 11 | M | 76 | A | 4th | Lives with spouse | 110 | Total gastrectomy | Laparoscopic | Yes | Chemotherapy |
| 12 | F | 82 | A | 4th | Lives alone | 62 | Partial gastrectomy | Open surgery | No | Nil |
| 13 | F | 59 | A | 2nd | Lives with spouse | 19 | Partial gastrectomy | Open surgery | No | Chemotherapy |
| 14 | M | 70 | B | 1st | Lives alone | 11 | Partial gastrectomy | Open surgery | No | Nil |
| 15 | F | 56 | M | 5th | Lives with parent | 33 | Total gastrectomy | Open surgery | Yes | Chemotherapy |
| 16 | F | 84 | A | 1st | Lives alone | 17 | Partial gastrectomy | Laparoscopic | Yes | Nil |
| 17 | M | 48 | A | 4th | Lives with parent | 9 | Total gastrectomy | Laparoscopic | Yes | Chemotherapy |
| 18 | M | 77 | A | 4th | Lives with spouse | 78 | Total gastrectomy | Open surgery | Yes | Nil |
| 19 | Fe | 58 | A | 3rd | Lives with spouse | 11 | Partial gastrectomy | Laparoscopic | No | Nil |
| 20 | M | 54 | A | 1st | Lives with spouse | 48 | Partial gastrectomy | Open surgery | No | Chemotherapy |

*A=, B=, C=, M=.
†Social deprivation quintile: 1st quintile being the least deprived, 5th quintile being the most deprived.

The study cohort included participants who had undergone surgery between 5 months and 14 years prior to the interviews. There did not seem to be a relationship between the length of time out of surgery and concerns about cancer recurrence.

## Theme 2: technical aspects of surgery

Several outcomes related to this theme were discussed by participants. Most importantly, participants focused on whether the surgical team was able to excise the 'cancer' in its entirety. This priority was often referenced in relation to the 'success' of surgery and its contribution to 'curing' participants of cancer:

> …thinking to yourself that, you know, everything has been done to the best of the hospital's ability, and, you know, they've taken absolutely everything out. (Participant 4)

While participants mostly referred to the cancer as a single 'entity', there were a small number who demonstrated some knowledge of the importance of different aspects of surgery such as lymph node excision:

> And yeah, I remember the news about the pathology on the bits they'd taken away, and the lymph node system and what not, came a week or two before I was due to go back on the chemo. (Participant 20)

Six participants underwent a minimally invasive surgery with the remainder undergoing open surgery. The size of the wounds or type of surgical approach was referred to by only a minority of our participants. In the main, these were made in passing as little importance was placed on the surgical approach:

> Interviewer: "Okay, and what…what did that mean for you to have keyhole surgery?"
>
> Respondent: "It didn't mean anything really, you know, I had…I'd heard about keyhole and people who'd had it." (Participant 5)

## Theme 3: adverse events following surgery

Eleven participants suffered a complication following surgery; however, this theme was important to all interviewees. Perioperative death was the most frequently discussed surgical complication:

> …the fear of dying on the operating table is really real. (Participant 2)

During their surgical consultation, participants retained some understanding of the risk of perioperative death and many were able to quote figures about how likely this complication was. Other complications were highlighted when recounting a personal experience. While all complications occurred in the postoperative period, there were several different causes attributed to these events (eg, direct surgical, anaesthesia-related and medication-related). The severity and consequences of the complications also varied significantly; some were

self-limiting and resulted in a minor extension of the length of hospital stay:

> But I was out of it for three days, I was just hallucinating and God knows what, probably because of the morphine. (Participant 7)

And those which were life-threatening and required significant clinical intervention:

> It was a twisted bowel. Yeah, I was told it was a twisted bowel. Because I always remember that when they brought me back from obviously having a look and everything, I always remember [they] said…we're going to have to take you back to surgery. (Participant 15)

The severity of the complications suffered did not seem to shape the key priorities in relation to participants' 'worst side effect' of surgery or 'most important outcome'; these almost entirely related to 'long-term impacts of surgery' and 'cure' respectively (see below) regardless of how long ago their operation was and which surgical approach (laparoscopic or open surgery) was employed.

While participants recounted that some of the more serious complications (eg, death, anastomotic leak and cardiopulmonary complications) were described by surgeons during the consent process, some were exposed to other sources of information in the preoperative stage. Participants were regularly provided with written information about their cancer and its management; however the quality and content of this varied depending on the location of their hospital. The response to this format was varied:

> I'm going to be honest with you, I didn't actually read them… because I didn't want things going in my head that I couldn't take in. (Participant 17)

Participants that read the written information often found them difficult to digest for several reasons including the volume of information and fear of the gravity of the diagnosis or prognosis:

> Well some of it were just waste of time, but others, you know, if you've got a book about that thick and you read through it and half of it applies to you, and the others just sort…doesn't apply, you know. (Participant 5)
>
> As much as it's alright handing leaflets out, I can…I am a bit of a reader so I will read stuff, but when you think you've got a death sentence you think, what's the point in reading that? (Participant 13)

Several hospitals had patient-support groups which provide a 'buddy' system for those awaiting surgery. These support groups became an important part of the recovery process and continue to be relevant many years after surgery. Patients found these more useful as they had the opportunity to speak to those with lived experience of the diagnosis and treatment. While this served

as an important source of information to tackle the longer-term impacts of surgery (below), these groups also provided comfort to patients:

> I think the support group and speaking to people that have been through it, because it can demystify it quite a lot. (Participant 2)

Peer support also provided additional sources of information to participants in the preoperative period with respect to some of the complications that could arise:

> And I know one guy, where the oesophagus junction was, he'd had that leaking, and he couldn't eat more than, like, grains of rice and things; so that would be pretty horrendous. (Participant 4)

As a result of verbal and written information from healthcare professionals and additional peer support, participants were able to describe key adverse without necessarily having experienced them firsthand.

### Theme 4: recovery from surgery

Experiences during the immediate postoperative recovery period were referenced by 18 study participants. While some participants' experience of recovery from surgery was directly linked to complications, there were aspects of recovery such as postoperative pain, mobility and the recommencement oral intake that were common among all those who spoke this theme.

Most participants did not mention postoperative pain as an important focus. Those that did, expected to suffer a degree of pain; however experiences of its severity varied widely. Pain levels among interviewees who had undergone similar operations through laparotomy incisions were not uniform. One participant who had open surgery described:

> I do remember waking up and really being in a hell of a lot of pain and being really out of it. (Participant 6)

While another who had undergone their surgery using the same approach recounted:

> I didn't really have much pain. (Participant 14)

Discussions around postoperative pain were not confined to discomfort from the surgical incisions, but also related to postoperative complications:

> I was back in writhing in agony with a serious infection in the wound. (Participant 9)

Participants recounted the limitations in their mobility during the postoperative period. There were many factors contributing to this, including physical weakness, not receiving appropriate encouragement to mobilise and being restricted by surgical drains:

> …really difficult to be mobile I suppose, and move around, yourself, 'cause obviously you've got quite a lot of tubes and different things coming out. I felt very, very swollen. (Participant 6)

### Theme 5: long-term problems following surgery

All participants described significant long-term symptoms related to surgery. For the most, this represented the 'worst side effect' in relation to their treatment, and outcomes from this theme were referenced more than any other theme further emphasising its importance. All participants described experience with struggling to eat and drink following surgery and the majority (16/20) talked extensively about the impact of fatigue on their daily lives. Problems with maintaining weight, issues with ongoing gastrointestinal symptoms and chronic pain were discussed by most participants.

Fatigue was described in many ways; 'exhaustion', 'feeling tired all the time', 'feeling so weak' and 'having no energy'. For the main, fatigue was a symptom which persisted for months after surgery and could impact on a participant's ability to undertake day-to-day activities or to socialise:

> Well, I'm so weak, I used to go out, you know, and do fishing and do things with my lads. I'm just getting that little bit better now after 8 months, but I'm so weak and tired. (Participant 18)

Adapting to fatigue was and for many continued to be a difficult challenge; however many participants understood that this was s a recognised and acceptable symptom to them given the magnitude of the surgery:

> I've come out with…more…appreciation for looking after myself and my…And if I'm tired, I stop. (Participant 15)

There were several causes for the struggles participants associated with eating and drinking. Participants often described having to eat and drink smaller volumes more frequently and some were unable to tolerate certain food types or consistencies. This had a direct effect on the pleasure associated with eating and an impact on where participants could eat:

> Well I don't eat what I would like to…But I know that for the rest of my life, I won't be able to go out for big meals, to big venues and eat like I used to eat before, you know. (Participant 5)

Most participants recounted being told prior to their surgery that their diet would be different and that they would have to 'learn how to eat again'. Despite this, some participants felt that not enough information was given to highlight the true impact of this long-term issue and methods to address it:

> I think it's a lot worse than what they tell you. Because like some days, I'll eat a certain thing which I've ate before, and you just can't breathe properly, it's choking you. (Participant 18)

A broad range of gastrointestinal symptoms were reported by participants. The timeframe relating to how long these persisted was similarly broad (sometimes months and years) and did not seem to follow a pattern.

Nausea, vomiting, diarrhoea, reflux and belching were the the most common problems described. Many of these symptoms resulted in significant impacts on quality of life (see theme 6):

> …I still get the bile reflux and I get this constant pain in the oesophagus which affects my sleeping as well. (Participant 7)

> The dumping syndrome was mentioned. Never understood it until it happened. You know, how my body reacted to certain foods that I'd normally eat that it doesn't like anymore. (Participant 15)

### Theme 6: long-term life impact of surgery

The long-term effects on 'normality', quality of life and psychological impact of surgery were discussed extensively by all patients. A strong desire to return to a form of 'normality' was regularly expressed. While the reference point for 'normality' differed among patients, common characteristics existed; namely, a desire to do what they used to do such as working, exercising, socialising with friends and family, and being able to travel:

> It's about living as I did before, and forgetting what had happened, and I do that quite often. (Participant 6)

The experience of returning to normality varied among those interviewed. Many participants were largely able to return to their 'normal' activities although with some modifications:

> Yes. I want to go on holidays again. I love cruises and I want…but until my eating's improved, I wouldn't do that. (Participant 1)

> Now that it's 18 months on, I am back to having what would be a normal life again, now, albeit with smaller portions of meals and things. (Participant 4)

Some participants, however, have not been able to return to activities that provided them with significant enjoyment:

> I've never actually got back to my normal activity. I've never played golf since that day and I used to love golf. (Participant 4)

In general, participants understood that life after gastrectomy would be different:

> I'm still alive, and then I need to get back to normal. It takes a while for you to realise your new normality is not like your old normality. (Participant 2)

While much of the discussion relating to 'normality' centred around specific tasks which participants valued or missed, the impact of gastrectomy on a participant's overall general quality of life was important to many. Many understood that quality of life needed to be redefined in comparison to life before surgery, but nonetheless there was a minimum level that would need to be achieved:

> If I have some sort of quality of life, where I can get up and wash myself and do, that is something that I'd live for. But I couldn't be sat there and nursed 24/7. (Participant 17)

The psychological burden on participants following gastrectomy is a significant one. Each of the previously discussed themes could impact on a participant's mental state, and while certain phases of the treatment pathway were time-limited, the psychological effects could persist for much longer;

> You don't just suffer from physical; you suffer from mental. And I think the mental is a lot more powerful that the physical, because you can shut pain off by taking medication, but it's very hard to shut problems off mentally. (Participant 18)

> You know, 'cause psychologically you think you've still got this poison in your body, as much as I've got rid of, you know, my monster. (Participant 13)

Some of the psychological impacts were associated with participants having to adapt to a new normality in relation to what they were able to do, what they were able to eat, how they looked physically or how they felt around others:

> Well, it was a problem because like I say, I've always been a proud chap and proud of my body because I kept myself fit and everything. When I looked in the mirror, quite distressing. That was it, yeah. It makes you feel inferior. (Participant 18)

> I feel a freak, I feel when I go into a big room with people that everybody has got a stomach and I haven't got one, it's not that I want them to know, but I just don't feel the same anymore… (Participant 8)

### Definition of 'outcome' by patients

All participants were asked what their understanding of the term 'outcome' was in the context of clinical research. Two participants were able to provide a broad-ranging definition which encompassed some of the benefits and adverse effects of treatment:

> My perception of what would be meant by that phrase would…at a variety of levels; it could be does the patient live or die? Does the patient recover to an acceptable state for an extended period of time, and my understanding of what that might be, would be a, sort of, five year period… (Participant 20)

One person stated that they did not know how to define the term, while the remainder defined 'outcome' by recounting a single outcome, which was most important to them:

> Okay, my understanding is that at the…the outcome would be that the cancer would be possibly all gone. (Participant 19)

Participants were asked to provide a single outcome that was 'most important to them'. Fifteen participants identified that the most important outcome was that they were 'cured of cancer', with the remaining five describing outcomes related to 'returning to normal' and being able to enjoy a 'good quality of life'. These priorities did not alter with respect to how long-ago surgery was performed, which approach was undertaken or how old the patient was.

## DISCUSSION

To our knowledge, this is the first in-depth qualitative study exploring the priorities of patients following potentially curative surgery for gastric cancer. The study focused on this cohort as one of the main aims was to identify outcomes which would be considered for inclusion in a COS for potentially curative surgical trials in gastric cancer. This present study will be used to help generate a list of outcomes that will be presented for prioritisation to healthcare professionals and patients in an international, multilanguage online Delphi survey. The results of the Delphi survey will inform a consensus meeting to finalise the COS. While other COS studies in the field of cancer[17–19] may have identified similar important outcomes such as survival, it was important to consider our participants separately given the unique problems which arise with gastrectomy. These include distinctive short-term and long-term problems related to surgery such as anastomotic leak and reactive hypoglycaemia (dumping syndrome) which would not be relevant in other COS.

The themes identified highlight the profound and wide-ranging physical, social and psychological impacts that gastrectomy has on patients which can persist for months and years. We have previously described the reporting of outcomes in surgical trials for gastric cancer over a 20-year period.[5] Most surgical trials in this field have focused on reporting short-term postoperative outcomes. While these are important to patients, they are not representative of the whole picture. This work highlights how patient priorities for outcomes may differ from the traditional surgical focus. More work is now needed to develop the COS which incorporates views of all key stakeholders including patients.

More than half of the 'top-ten' most frequently discussed outcomes in our study related to longer-term issues such as problems with eating, returning to 'normality', fatigue, weight loss, gastrointestinal symptoms and psychological impacts. These types of outcomes are infrequently reported in surgical trials and demonstrate that researchers within this field have not reflected the priorities of patients. This challenge needs to be addressed using an approach which is inclusive of patients and their views.

The GASTROS study aims to develop a COS; critically important outcomes which should be reported—as a minimum—by future surgical trials for gastric cancer.[6] By standardising the reporting of such outcomes, it aims

to improve the ability to synthesise evidence, reduce research waste and ultimately aid researchers in answering important questions related to gastrectomy. The first stage in developing the COS consists of identifying a 'long-list' of outcomes which will then be prioritised by key stakeholders during an international online Delphi survey. The process of developing the long-list should be comprehensive and involve both healthcare professionals and patients in order to minimise the risk of omitting potentially important outcomes. Our study reaffirms the importance of a mixed-methods approach to identifying potentially important outcomes. As other COS developers have found, building a long-list based solely on outcomes reported in previous trials or as developed by clinicians often neglects the views of key stakeholders.[8–10] This ultimately runs the risk of producing a COS which does not reflect the priorities of patients which does little to address the current challenges with outcome reporting.

Understanding patient priorities following gastrectomy is invaluable for other reasons. Patients with gastric cancer want detailed information about their condition and treatment.[20] With the knowledge that long-term impacts of surgery are important, healthcare professionals can tailor the consent process prior to surgery to ensure that the patient has a better understanding of these and is making an informed decision. Considering patient priorities may also have implications for the future development of national and international audits.[3 4] For several pragmatic reasons, most comprehensive gastric cancer surgery audits focus on short-term outcomes. Identifying methods to report longer-term quality outcome measures may make such audits more relevant to patients. Studies assessing patients' views in similar disease areas found similar things, that long-term outcomes (survival and long term quality of life) were important.[21]

### Strengths and weaknesses

The study was able to gain an in-depth understanding of patient priorities based on the experience of participants with a broad range of characteristics representative of those undergoing surgery for gastric cancer in the UK.[4] Furthermore, our purposive sampling approach was established a priori in a study protocol which had undergone a robust peer-review process.

This study was also able to highlight and address significant challenges associated with the comprehension of medical language by patients; particularly terms central to the development of a COS. Patients largely did not understand the use of the term 'outcome' within the context of medical research. Once it was defined as an 'impact or effect of a treatment which may be beneficial or harmful', participants were more easily able to describe their key priorities in outcome reporting for future trials. This has several implications for the GASTROS study as well as other COS projects moving forward. It highlights the importance of ensuring that the premise of the study is clear and understood by all participants, especially patients; outcomes included in the Delphi survey must be

presented and explained in a manner which is accessible to all; 'outcome' must be clear when adapting it to other regions where there may be no direct translation for the term.

The interviews were conducted by an expert in the field of gastric cancer surgery which may have resulted in a degree of observer bias. To mitigate this potential limitation, the study management team (which was made up primarily of members unfamiliar with gastric cancer surgery) was involved in ongoing discussions during data collection and analysis. It is also possible that patients modified their responses because of awareness of the background of the interviewer. Every effort was made to follow the semistructured interview schedule, to put the patients at ease and take time to let them talk. The average length of the interviews (greater than 45 min) reflects the time patients were given to express their views.

A further potential limitation of this present interview study is a lack of international patient participation. Consequently, there may be outcomes which are relevant to non-UK patients that have not been identified. Gastric cancer is an international disease, and cultural and regional influences may alter expectations and priorities of patients. While we have not identified evidence from COS developers in other fields that confirms these variations, it remains a possibility. Our reasons for limiting the interviews to UK-only patients were primarily down to pragmatism and finite resources. To mitigate this, the Delphi survey will be available in several languages and during the first round, all participants will be able to submit additional outcomes that they believe were omitted. These will be considered by the study team and presented for prioritisation by participants in round two of the Delphi survey, if appropriate. Adopting this approach also enables the exploration of regional variations in outcome priorities which may form the basis of a future international qualitative study.

This study focuses primarily on the impact of gastrectomy from the perspective of patients. However, we acknowledge that major complex surgery such as gastrectomy inevitably results in both direct and indirect effects on family members and caregivers. While these wider impacts warrant further examination, we limited participation in this present study to patients, as the scope of the COS aims to consider the perspective and priorities of patients, surgeons and oncology nurses. Part of our planned future work is to review the COS to ensure that it remains up-to-date and relevant. At this point, it will be possible to widen participation beyond these three groups to include caregivers and other allied healthcare professionals.

Recruitment to the study stopped when no new data with interpretative value were identified.[22] However, 'data saturation' is a topic which deserves further discussion as there is no way of knowing for certain that no new outcomes would have been identified had further interviews been undertaken. Some argue that the term 'data saturation' is often misused and misunderstood

and should be operationalised in a way consistent with the scope of the study being undertaken.[23] As described above, the ability of patients and healthcare participants to suggest further outcomes in round one of the Delphi survey aims to mitigate against this potential limitation.

Most participants had undergone their surgery at least 12 months prior to this study. As such, it should be acknowledged that there may have been a greater exploration of and emphasis on shorter-term outcomes had we recruited more participants from a shorter postoperative time period. Again, to address this potential limitation, we plan to recruit participants for the Delphi survey from all postoperative periods and will have the opportunity to examine whether 'time from surgery' affects patient priorities.

In summary, this study identified 38 unique outcomes which are important to patients following surgery for gastric cancer. Many of these outcomes are poorly represented by trials within this research field. These outcomes will be added to other potentially important outcomes to be considered for prioritisation by key stakeholders to develop a COS for surgical trials in gastric cancer.

**Author affiliations**
[1]Department of Oesophago-Gastric Surgery, Salford Royal NHS Foundation Trust, Salford, UK
[2]Division of Cancer Sciences, School of Medical Sciences, Faculty of Biology, Medicine and Health, The University of Manchester, Manchester, UK
[3]Centre for Surgical Research and Bristol NIHR Biomedical Research Centre, University of Bristol, Bristol, UK
[4]Paediatric ENT Department, Manchester University NHS Foundation Trust, Manchester, UK
[5]Division of Infection, Immunity and Respiratory Medicine, Faculty of Biology, Medicine and Health, University of Manchester, Manchester, United Kingdom
[6]Centre for Primary Care, University of Manchester, Manchester, UK

**Acknowledgements** The authors would like to thank all those who participated in this study. The authors would also like to highlight the role undertaken by Dr Aleksandra Metryka, Senior Clinical Trials Coordinator, who facilitated the running of the study.

**Contributors** Conceptualisation: BA, JB, IB and RLM. Data curation: BA. Formal analysis: BA, RLM. Funding acquisition: BA, JB and IB. Investigation: BA and RLM. Methodology: BA, IB and RLM. Project administration: BA, IB, RLM. Resources: BA and IAB. Software: BA. Supervision: JB, IB and RLM. Validation: BA and RLM. Visualissation: BA, JB, IB, RLM. Writing (original draft): BA. Writing (review and editing): BA, JB, IB and RLM.

**Funding** This study is funded by the National Institute for Health Research (NIHR) Doctoral Research Fellowship Grant (DRF-2015-08-023). JB is partially funded by the NIHR Bristol Biomedical Research Centre.

**Disclaimer** This paper presents independent research funded by the National Institute for Health Research (NIHR). The views expressed are those of the author(s) and not necessarily those of the NHS, the NIHR or the Department of Health.

**Competing interests** None declared.

**Patient consent for publication** Not required.

**Ethics approval** The study was given ethical approval by the National Research Ethics Service North West—Cheshire (11/NW/0739) and governance approvals by Central Manchester University Hospital NHS Foundation Trust.

**Provenance and peer review** Not commissioned; externally peer reviewed.

**Data availability statement** Data are available on reasonable request.

**ORCID iDs**
Bilal Alkhaffaf http://orcid.org/0000-0001-5751-1846
Rebecca L Morris http://orcid.org/0000-0003-1587-0802

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
