## [Reviewer comments · BMJ Open]

ARTICLE DETAILS

TITLE (PROVISIONAL)	Patient Priorities in Relation to Surgery for Gastric Cancer: Qualitative Interviews with Gastric Cancer Surgery Patients to Inform the Development of a Core Outcome Set.
AUTHORS	Alkhaffaf, Bilal; Blazeby, Jane; Bruce, Iain; Morris, Rebecca L.

VERSION 1 – REVIEW

REVIEWER	Virginia Sun City of Hope, USA
REVIEW RETURNED	29-Oct-2019

GENERAL COMMENTS	This is a well-written manuscript that aims to explore the outcomes that are relevant to patients undergoing gastric surgery. I have the following comments for the authors to consider:  1. The detailed eligibility criteria table (table 2) notes patients who has surgery for palliative intent were excluded. What was the rationale for this exclusion? Patient-centered outcomes are equally important in palliative surgery populations as curative intent surgery. 2. Data analysis: it appears that only two investigators were involved with qualitative data analysis. If there were disagreements on coding, how were the differences resolved? Did a third investigator participate in adjudication?
---

REVIEWER	Leah Gramlich University of Alberta
REVIEW RETURNED	04-Nov-2019

GENERAL COMMENTS	Thanks for this important work. The inclusion of patients who are undergoing surgery for curative intent however, limits the generalizability of the findings given the nature of gastric ca and the fact that it often presents late. - I think this warrants further discussion and rationalization - for instance, at surgery for curative intent, it may be discovered that disease has spread and it is not resectable - however, that patient has expectations as they go into surgery. In addition, i wonder if there would be value in considering things from the patient perspective - specifically, the signals related to long term problems following surgery are as prominently reflected in the numbers as are issues related to survival - these are things that are impacting patients on a daily basis - issues with eating/nutrition, GI symptoms and fatigue - this study is nested in a larger surgical study and many of the themes relate to that - can patient symptoms and concerns not be the priority? (i wonder if you are looking through the provider and not the patient lens.
---

	There is no framework utilized to drive patient oriented care considerations such as the IAP2 framework - it would be of value to include consideration of this to better inform how patient perspectives are included going forward in this work.
--	--

REVIEWER	Stefanie Deckert Center for Evidence-based Healthcare; University Hospital Dresden; Germany
REVIEW RETURNED	08-Nov-2019

GENERAL COMMENTS	The study addresses patients' perceptions of gastric cancer patients undergoing surgery. This well written and structured manuscript raises interesting issues regarding what patients deem as relevant in terms of outcomes. Some questions deserve further attention, however. In general: There are other COS initiatives and qualitative studies focusing on other entities and/or surgery. Authors should explain and discuss why these research results are not applicable/usable to the population described here (cf. discussion about research waste). Data saturation is an important quality criterion of qualitative analysis. Even though, "further outcomes were identified from the following two transcripts" it cannot be ruled out that further interviews (e.g. #21) would have yielded further patient-relevant outcomes. Authors should discuss this point more critically. Minor points: Abstract/ Introduction - the three terms „priorities, outcomes and themes“ are confusing. What is meant with priorities and themes compared to outcomes? Abstract - „thematic analysis“ needs to be described in more detail (e.g. grounded theory) Methods: - Page 11/line 14: reads very cryptically and needs further explanations and examples Table 1: - Question 10: what was your intention that patients should define the word outcome themselves
---

REVIEWER	J. Park University of Manitoba
REVIEW RETURNED	18-Nov-2019

GENERAL COMMENTS	Thank you for the opportunity to review this paper. I think it is very well designed, conducted, and written. I think the methods were particularly well done and well described. I have only a couple of comments: 1. The authors tried to use a purposeful sampling strategy - including patients < 1 year from surgery, 1-3 years from surgery, or > 3 years. But there were only 3 participants that had had surgery within the last year, one of whom was at the 11 month point. This is notable, since responses reflect participant's perceptions and their recollection of events, and if a lot of time has passed, their responses may be affected by their recall. This may be one of the reasons that the descriptions of their initially
--

	consultation were "sketchy" 16. This may also help explain why there wasn't much of a focus on pain during the recovery - maybe they don't recall all the events that well anymore. I think it is best to conduct the interview as close as possible to when you want to know what their perceptions are. If the immediate post-op or shorter term outcomes are of interest, then that's when the interviews should occur (if possible). As such, with when the interviews occurred, the concern is that this reflects more on intermediate to longer term "outcomes" for participants, with some recollection of the shorter term post-surgery outcomes. I think at the very least this should be acknowledged as a limitation. 2. Were all the patients NED (no evidence of disease)? This information would be useful for readers as well for several reasons. First, it can affect participant's perspectives. Once a recurrence happens, their reported quality of life decreases and they may have a higher risk of other emotional changes like depression. Second, it's important to know because otherwise, the responses reflect the perceptions of disease free survivors, and we don't really know if the perceptions of people who recur are the same.
--	---

VERSION 1 – AUTHOR RESPONSE

Reviewer: 1

Reviewer Name: Virginia Sun

Institution and Country: City of Hope, USA

Please state any competing interests or state 'None declared': None declared

Please leave your comments for the authors below

This is a well-written manuscript that aims to explore the outcomes that are relevant to patients undergoing gastric surgery. I have the following comments for the authors to consider:

1. The detailed eligibility criteria table (table 2) notes patients who has surgery for palliative intent were excluded. What was the rationale for this exclusion? Patient-centered outcomes are equally important in palliative surgery populations as curative intent surgery.

This study should be seen in the context of a larger project to develop a core outcome set (COS) for potentially curative surgical trials in gastric cancer. Whilst we agree the views of patients undergoing palliative interventions are important, the scope of this COS is for curative treatment, hence why palliative patients were excluded. We have added this to the discussion for clarification.

2. Data analysis: it appears that only two investigators were involved with qualitative data analysis. If there were disagreements on coding, how were the differences resolved? Did a third investigator participate in adjudication?

There were no disagreements about coding, but had there been, these would have been discussed with the study management group. Themes were agreed by all authors and this has now been added to the manuscript.

Reviewer: 2

Reviewer Name: Leah Gramlich

Institution and Country: University of Alberta

Please state any competing interests or state 'None declared': none declared

Please leave your comments for the authors below Thanks for this important work.

The inclusion of patients who are undergoing surgery for curative intent however, limits the generalizability of the findings given the nature of gastric ca and the fact that it often presents late. - I think this warrants further discussion and rationalization - for instance, at surgery for curative intent, it may be discovered that disease has spread and it is not resectable - however, that patient has expectations as they go into surgery.

This study should be seen in the context of a larger project to develop a core outcome set (COS) for surgical trials in gastric cancer. The scope of the COS is for potentially curative treatment and hence why non-curative patients were excluded. We have added this to the discussion for clarification. The issue of resectability of the tumour at surgery was covered by the theme 'technical aspects of surgery' which we expand on in appendix 1.

In addition, i wonder if there would be value in considering things from the patient perspective - specifically, the signals related to long term problems following surgery are as prominently reflected in the numbers as are issues related to survival - these are things that are impacting patients on a daily basis - issues with eating/nutrition, GI symptoms and fatigue - this study is nested in a larger

surgical study and many of the themes relate to that - can patient symptoms and concerns not be the priority? (i wonder if you are looking through the provider and not the patient lens.

We have not attempted to place priority/importance of one theme or outcome over another in this study. The symptoms that are referred to by the reviewer have been considered and will be presented to participants in the next stage of the wider GASTROS study. The study gives equal weight to both healthcare professionals and patients in the identifying core outcomes to report in future trials.

There is no framework utilized to drive patient oriented care considerations such as the IAP2 framework - it would be of value to include consideration of this to better inform how patient perspectives are included going forward in this work.

This study is funded by the UK's National Institute of Health Research. We use the NIHR's INVOLVE standards for all patient and public participation. This has been acknowledged in the methods section and referenced accordingly.

Reviewer: 3

Reviewer Name: Stefanie Deckert

Institution and Country: Center for Evidence-based Healthcare; University Hospital Dresden; Germany

Please state any competing interests or state 'None declared': None declared

Please leave your comments for the authors below The study addresses patients' perceptions of gastric cancer patients undergoing surgery. This well written and structured manuscript raises interesting issues regarding what patients deem as relevant in terms of outcomes. Some questions deserve further attention, however.

In general:

There are other COS initiatives and qualitative studies focusing on other entities and/or surgery. Authors should explain and discuss why these research results are not applicable/usable to the population described here (cf. discussion about research waste).

As with much research using similar methodologies, there may be overlap. It may be presumed that all cancer patients undergoing curative treatments would prioritise survival and cure as an outcome. However, gastric cancer and its treatment (an outcomes of treatments) is unique in the same way that other pathologies and treatments are unique. For example, outcomes related specifically to gastrectomy such as post-operative reactive hypoglycaemia/Dumping syndrome or anastomotic leak would not be seen for example in surgical treatments for prostate cancer or chemoradiotherapy for anal cancer (both of which have developed COS). This study should be seen in the context of a larger project to develop a core outcome set (COS) for surgical trials in gastric cancer, which is why we need a gastric cancer specific qualitative study to identify priorities relevant to gastrectomy. We have added this to the beginning of the discussion section for clarification. We would also like to state that moving forward, we have already planned collaboration to identify appropriate measurement instruments for outcomes shared across different COS.

Data saturation is an important quality criterion of qualitative analysis. Even though, “further outcomes were identified from the following two transcripts“it cannot be ruled out that further interviews (e.g. #21) would have yielded further patient-relevant outcomes. Authors should discussed this point more critically.

We have now addressed this issue more critically in the limitations section of the discussion. We have referenced a recent publication by Saunders et al. examining the topic of ‘data saturation’ and their recommendation that this should be linked to the scope of the study. We have described how we plan to mitigate for outcomes not identified from these interviews in the next stage of the wider GASTROS study.

Minor points:

Abstract/ Introduction

- the three terms „priorities, outcomes and themes“ are confusing. What is meant with priorities and themes compared to outcomes?

This has been changed to 'outcomes'.

Abstract

- „thematic analysis“ needs to be described in more detail (e.g. grounded theory)

This has now been expanded on in the methods section.

Methods:

- Page 11/line 14: reads very cryptically and needs further explanations and examples

This has been removed and an explanation of the approach to disagreement has been described.

Table 1:

- Question 10: what was your intention that patients should define the word outcome themselves

Given the term 'outcome' is central to the wider GASTROS study, our intention was to understand whether participants could understand this term without further explanation. The resulting responses demonstrated the importance of presenting key research concepts to lay participants using language which is easy to understand from their perspective.

Reviewer: 4

Reviewer Name: Jason Park

Institution and Country: University of Manitoba

Please state any competing interests or state 'None declared': None declared

Please leave your comments for the authors below Thank you for the opportunity to review this paper.

I think it is very well designed, conducted, and written. I think the methods were particularly well done and well described. I have only a couple of comments:

1. The authors tried to use a purposeful sampling strategy - including patients < 1 year from surgery, 1-3 years form surgery, or > 3 years. But there were only 3 participants that had had surgery within the last year, one of whom was at the 11 month point. This is notable, since responses reflect participant's perceptions and their recollection of events, and if a lot of time has passed, their responses may be affected by their recall. This may be one of the reasons that the descriptions of their initially consultation were "sketchy" 16. This may also help explain why there wasn't much of a

focus on pain during the recovery - maybe they don't recall all the events that well anymore. I think it is best to conduct the interview as close as possible to when you want to know what their perceptions are. If the immediate post-op or shorter term outcomes are of interest, then that's when the interviews should occur (if possible). As such, with when the interviews occurred, the concern is that this reflects more on intermediate to longer term "outcomes" for participants, with some recollection of the shorter term post-surgery outcomes. I think at the very least this should be acknowledged as a limitation.

We have now acknowledged this in the discussion along with how we aim to address this in future stages of the wider study.

2. Were all the patients NED (no evidence of disease)? This information would be useful for readers as well for several reasons. First, it can affect participant's perspectives. Once a recurrence happens, their reported quality of life decreases and they may have a higher risk of other emotional changes like depression. Second, it's important to know because otherwise, the responses reflect the perceptions of disease free survivors, and we don't really know if the perceptions of people who recur are the same.

This is an important comment. None of our participants had evidence of recurrence at the time of interview. We are aware that at least one participant has since been diagnosed with recurrence (but was unaware at the time) and another was being investigated for suspected recurrence at the time of interview. We have highlighted this point in the results section. Once again, we would like to stress that this study's pre-defined scope was to identify outcomes that may be considered as part of a COS and hence exploring other areas, whilst important, may be best addressed by further research.

VERSION 2 – REVIEW

REVIEWER	Virginia Sun City of Hope, USA
REVIEW RETURNED	20-Dec-2019

GENERAL COMMENTS	I commend the authors for their work on making surgical research more patient-centered through the rigorous identification of core outcomes most relevant to patients in gastric/upper GI surgery. I was wondering if the authors can provide a minor comment in the discussion section on whether the development of COS should also include informal/family caregiver perspectives. The focus has traditionally been on patients (and rightly so), but for some cancer
--

	populations, including gastric, many of the unique postop needs often involve informal/family caregivers as well. The unique eating problems faced by this population requires adjustments in meal-related behaviors; this may also impact informal caregiver/family caregivers as many may be grocery shopping or preparing meals. Thank you.
--	--

REVIEWER	Stefanie Deckert Center for evidence-based healthcare; University Hospital Dresden, Germany
REVIEW RETURNED	17-Dec-2019

GENERAL COMMENTS	I agree with revised version. Many thanks.
--

REVIEWER	Jason Park University of Manitoba
REVIEW RETURNED	16-Dec-2019

GENERAL COMMENTS	My previously raised concerns have been addressed. I have no additional comments. Overall, I think the paper is well written.
---

VERSION 2 – AUTHOR RESPONSE

We would like to thank the reviewers for their additional comments. Please find below our response to reviewer 1's comment:

We have acknowledged that the perspectives of caregivers are important and that they should be considered. We have justified why we did not seek to recruit caregivers to this present study and how their perspective may be considered as part of our future work.

We hope that you will now consider our revised manuscript in an expedited manner and look forward to your response.